pharmacoepigenetics; pharmacoepigenomics; epigenetics; pharmacogenomics; molecular QTLs

**Author for correspondence:**
Russ B. Altman,
Email: Russ.Altman@stanford.edu

# Promises and challenges in pharmacoepigenetics

Delaney A. Smith[1] 🅘, Marie C. Sadler[1,2,3] 🅘 and Russ B. Altman[1] 🅘

[1]Department of Bioengineering, Stanford University, Stanford, CA, USA; [2]University Center for Primary Care and Public Health, Lausanne, Switzerland and [3]Swiss Institute of Bioinformatics, Lausanne, Switzerland

## Abstract

Pharmacogenetics, the study of how interindividual genetic differences affect drug response, does not explain all observed heritable variance in drug response. Epigenetic mechanisms, such as DNA methylation, and histone acetylation may account for some of the unexplained variances. Epigenetic mechanisms modulate gene expression and can be suitable drug targets and can impact the action of nonepigenetic drugs. Pharmacoepigenetics is the field that studies the relationship between epigenetic variability and drug response. Much of this research focuses on compounds targeting epigenetic mechanisms, called epigenetic drugs, which are used to treat cancers, immune disorders, and other diseases. Several studies also suggest an epigenetic role in classical drug response; however, we know little about this area. The amount of information correlating epigenetic biomarkers to molecular datasets has recently expanded due to technological advances, and novel computational approaches have emerged to better identify and predict epigenetic interactions. We propose that the relationship between epigenetics and classical drug response may be examined using data already available by (1) finding regions of epigenetic variance, (2) pinpointing key epigenetic biomarkers within these regions, and (3) mapping these biomarkers to a drug-response phenotype. This approach expands on existing knowledge to generate putative pharmacoepigenetic relationships, which can be tested experimentally. Epigenetic modifications are involved in disease and drug response. Therefore, understanding how epigenetic drivers impact the response to classical drugs is important for improving drug design and administration to better treat disease.

## Impact statement

Pharmacoepigenetics studies how epigenetic mechanisms impact disease states, and the response to drugs. We summarize work in these areas and propose an approach to move the field forward by using publicly available data to identify pharmacoepigenetic interactions. These interactions promise an improved understanding of how to deliver more precise personalized medicine.

## Introduction

Variation in genetics can lead to variation in the response to drugs. Pharmacogenetics (PGx) is the field of research that characterizes this relationship by examining how genetic variation correlates with pharmacological parameters such as pharmacokinetics and pharmacodynamics. In PGx, the simplified paradigm for understanding a gene–drug interaction is as follows: first, recognizing regions of genetic variation, second, identifying key functional changes such as single nucleotide polymorphisms (SNPs), and insertion and/or deletions (indels), and third, mapping these onto a drug response phenotype. This process represents a base case PGx scenario. However, there are cases where individuals have several variant genes of interest, called pharmacogenes. The effect from each pharmacogene complicates the analysis of how a single gene variant impacts drug response. PGx studies have been successful in explaining and predicting differences in drug response (Ross et al., 2012).

Genetic variation typically accounts for approximately 10–30% of observed differences in individual responses to drugs (Ross et al., 2012). Researchers have proposed that epigenetic effects, which modify gene expression without altering the genetic code, may also contribute to variation in drug response (Berger et al., 2009; Gomez and Ingelman-Sundberg, 2009; Kacevska et al., 2011; Ivanov et al., 2012; Ingelman-Sundberg et al., 2013; Kim et al., 2014; He et al., 2015; Stefanska and MacEwan, 2015; Cascorbi and Schwab, 2016). Epigenetic mechanisms include DNA methylation (DNAm), hydroxymethylation, histone modification, chromatin architecture changes, and noncoding RNAs (although RNAs are not always considered epigenetic factors) (Kelly et al., 2010). Epigenetic principles have been extensively studied in recent years, and a

thorough review of these principles in the context of health and disease is given by Zhang et al. (2020).

Epigenetic variants have been found near genes and gene regulators, which control the metabolism of drugs, suggesting a role for epigenetic mechanisms in modulating pharmacokinetics and pharmacodynamics (Kacevska et al., 2012; He et al., 2015; Shi et al., 2017). Pharmacoepigenetics, is the field that studies how epigenetic variability impacts variability in drug response. We can use a similar approach as with PGx to study this field. First, we identify variation in epigenetic markers, second, we select key epigenetic biomarker(s) in regions of variance, and third, we map these biomarker(s) to a drug response phenotype.

We introduce the term forward pharmacoepigenetics to describe situations where the existing epigenetic state dictates response to drugs (Csoka and Szyf, 2009; Cascorbi, 2013). However, as in PGx, there are more complex cases where epigenetics and drug response interact. Drugs can also modulate the epigenetic profile in a manner we call reverse pharmacoepigenetics, where compounds target epigenetic mechanisms to alter gene expression (Csoka and Szyf, 2009; Kelly et al., 2010; Lötsch et al., 2013; Ivanov et al., 2014; Figure 1). With some drugs, this is a side effect, but a subset of epigenetic drugs, which we will refer to as epi-drugs, are designed to alter epigenetic markers and are used to treat cancers, immune disorders, and mental health disorders (Peedicayil, 2014; Furtado et al., 2019; Licht and Bennett, 2021).

Biomarkers, such as methylation patterns, are used to identify whether an epi-drug will be effective for a specific patient (Treppendahl et al., 2014; Majchrzak-Celińska and Baer-Dubowska, 2017). Therefore, much of the literature in the field of pharmacoepigenetics analyzes biomarkers related to epi-drugs. One class of biomarker analyses identifies which patient cohorts respond best to certain epi-drugs (Berdasco and Esteller, 2019; Incorvaia et al., 2020). Another application seeks biomarkers that are indicative of epi-drug efficacy against a specific disease (Cheng et al., 2014; Alag, 2019; Morel et al., 2020). These studies have generated a large volume of data following advances in epigenetic sequencing technology (Zhou et al., 2015; Luo et al., 2020; Boix et al., 2021). The sequencing methods are reviewed in Cazaly et al. (2019). Computational methods to predict biomarkers and patient responses from the available information are emerging (Cazaly et al., 2019). For example, machine learning (ML)-based algorithms trained on functionally validated pharmacogenomic biomarkers joined with clinical measures, predicted selective serotonin reuptake inhibitor (SSRI) remission and response in patients with major depressive disorder (Athreya et al., 2019).

While epi-drug biomarker studies are important for improved patient diagnosis and treatment, they are not the focus of this review. Instead, we focus on the relationship between classical drug response and epigenetic variation (Gomez and Ingelman-Sundberg, 2009; Cascorbi, 2013; Lauschke et al., 2019). In PGx, researchers typically identify key SNPs from regions of genetic variance and map how they relate to the drug response phenotype. In pharmacoepigenetics, few studies have reported on how epigenetic variation relates to classical drug response. We find that (1) epigenetic variants can be indicative of disease and are varied throughout the population, (2) epigenetic variation can be summarized by key biomarkers, which predict diagnosis and prognosis, and (3) epigenetic variation impacts nonepigenetic drug response. We also demonstrate how publicly available data can be used to examine all aspects of the relationship between epigenetic variance and classical drug response to further our understanding of human biology and improve our treatment of disease.

## Epigenetic variation in disease, drugs, and drug response

### Epigenetic variation

Of the epigenetic mechanisms, DNA methylation (DNAm) is the most studied, and has been implicated in several disease phenotypes. DNAm at the promoter region of a gene is more likely to downregulate gene expression, while methylation in the body of the gene is more likely to increase expression (Jjingo et al., 2012). Cancers, immune diseases, and diabetic kidney disease are linked to differential methylation (Husquin et al., 2018; Kato and Natarajan, 2019; Ochoa-Rosales et al., 2020). Mental illnesses including schizophrenia, bipolar disorder, major depressive disorder, Alzheimer's, and autism may also be associated with variant methylation patterns (Tyrka et al., 2012; Cacabelos and Torrellas, 2015; Andari et al., 2020; Zhou et al., 2021). Based on these data, we focus the scope of our review on the relationship between DNAm and drug response. There are relationships between drug response and other epigenetic mechanisms, which are outlined in Kim et al. (2014), He et al. (2015), and Cascorbi and Schwab (2016).

Epigenetic variation relevant to disease can vary across ethnicities. Nielsen et al. (2010) found that DNA methylation of the l-opioid receptor gene (OPRM1) promoter region varied across African American, Hispanic, and Caucasian ethnic groups. This was done in the context of a heroin addiction study, and in addition to higher methylation levels in former heroin addicts compared to controls, and there was a significant difference across ethnicities in both users and control subjects. Epigenetic polymorphisms of the

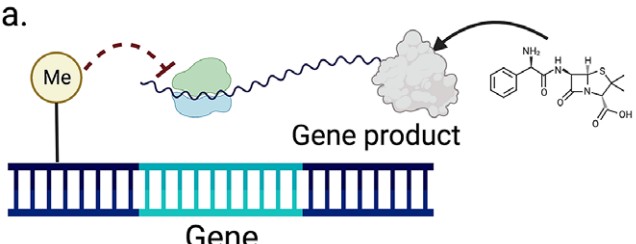
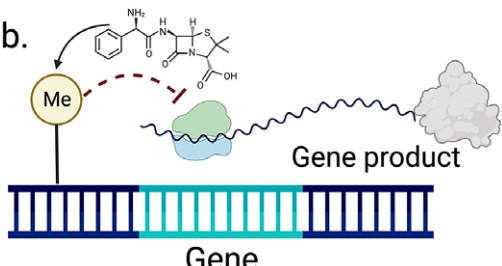

**Figure 1.** (a) Forward case of pharmacoepigenetics where basal DNAm influences drug response. Preexisting DNAm markers (Me) in the promoter region of the gene can downregulate gene expression. This decreases the amount of gene product available for interaction with a drug. Since the DNAm affects the drug response phenotype, we call this forward pharmacoepigenetics. (b) Reverse case where drug changes DNAm and this in turn affects response phenotype. Here the drug is altering the methylation status of the gene promoter region, which leads to changes in downstream gene expression. Since the drug is affecting the DNAm, we call this reverse pharmacoepigenetics. In this scenario methylation at the promoter region downregulates gene expression, but this is not always the case. Created with BioRender.com.

gene *CYP2D6* are present among Chinese Tibetan, Mongolian, Uygur, and Han populations (Qi et al., 2020). *CYP2D6* is important for susceptibility to cardiovascular disease and drug metabolism. These studies demonstrate that methylation patterns vary across different populations in genes relevant to disease phenotype and drug response.

### Key epigenetic biomarkers predict diagnosis and prognosis

For diseases with an epigenetic component, it is useful to identify key biomarkers representing differential epigenetic states associated with disease diagnosis, prognosis, and treatment efficacy. Duruisseaux et al. (2018) identified key DNAm biomarkers that were predictive of response to anti-programmed death 1 (anti-PD-1) treatment in nonsmall-cell lung cancer patients. In an epilepsy study, a methylation signature defined temporal lobe epilepsy and predicted drug resistance in patients (Xiao et al., 2020).

Computational models can aid in identifying epigenetic biomarkers, which correlate to disease. For example, three ML programs were trained on the sequence of DNA with CG repeats (CpG sites). Researchers used these three programs (a complexity-optimized classifier, a decision tree, and a miRNA expression-based decision tree) to predict paclitaxel-sensitive and resistant breast cancer tumors (Bomane et al., 2019). ML programs have also used drug similarity data and cancer cell similarity matrices to predict the sensitivity of various cancer cell lines to novel drugs. The performance of this ML prediction model using DNAm was comparable to that of experimentally based information from oncogene mutation and gene expression data (Yuan et al., 2020). These studies demonstrate that biomarkers represent epigenetic variance in a manner, which can be used to predict disease phenotype, prognosis, and treatment efficacy.

### Epigenetic variation impacts nonepigenetic drug response

Epigenetic profiles vary in the population, are associated with several diseases, and can be represented by biomarkers to identify and predict disease phenotypes. Epigenetic alterations can also influence the response to nonepigenetic drugs. For example, methylation patterns in patients with Fragile X syndrome correlate with a differential response to a mGluR5 antagonist, which may alleviate some symptoms of the disorder (Jacquemont et al., 2011). In another hereditary disease, hyperhomocysteinemia, there is a consistent association between methylation levels of *Betaine–Homocysteine S-Methyltransferase* and folate therapy efficacy (Li et al., 2019).

Differences in methylation status correlate with the presence and severity of several mental health disorders. Methylation status also impacts response to anti-psychotic treatment (Swathy et al., 2018). One example of this relationship is that DNA methylation status in the interleukin-11 gene predicts clinical response to anti-depressants (Powell et al., 2013). Another study done on the Chinese Han population found that response to the anti-psychotic risperidone could be explained by both SNPs in key genes and CpG islands in the promoter or gene coding regions of those genes (Shi et al., 2017). Specifically, examining methylation in *CYP3A4, CYP2D6, ABCB1, HTR2A,* and *DRD2* genes revealed seven significant CpG sites within the promoter or coding regions of these genes. Zhou et al. (2021) expand on the correlative relationship between DNAm profiles and clinical response to antipsychotic drugs in a retrospective study.

Most of the genes identified in these studies are involved in drug metabolism. Absorption, distribution, metabolism, and excretion (ADME) genes often vary in expression level. This variation impacts drug pharmacokinetics. Cytochrome P450 (*CYP*) genes are ADME genes and there is evidence that some unexplained variance in their level of function is attributable to methylation (Kacevska et al., 2012; Shi et al., 2017; Xiong et al., 2022). Polymorphisms in *CYP450* and another gene, *ABCB1,* and their respective DNA methylation statuses significantly altered risk for steroid-induced osteonecrosis in the femoral head in Chinese populations (Huang et al., 2020). Moreover, methylation of *ABCB1* also had a significant effect on aspirin resistance in Chinese ischemic stroke patients (Xu and Wang, 2022). Furthermore, methylation of *CYP1A1* was shown to modulate stable warfarin dosage in Chinese patients (He et al., 2021).

*CYP3A4* is another cytochrome P450 family member that exhibits high interindividual variation in hepatic expression. Much of the variability in *CYP3A4* remains unexplained. However, there exist highly variable CpG methylation sites in adult livers, which correspond to important *CYP3A4* transcription factor binding sites at the proximal promoter. This suggests that the variance in the expression of *CYP3A4* in adult livers may be due to methylation of the proximal promoter region (Kacevska et al., 2012).

### Epigenetic and pharmacogenomic resources

Several publicly available resources provide useful epigenetic and pharmacogenomic information. These resources derive from initiatives to aggregate molecular association studies to create data resources that are publicly available (The GTEX Consortium, 2020; Min et al., 2021; Battram et al., 2022; Ruiz-Arenas et al., 2022; Xiong et al., 2022). We focus on DNAm as an epigenetic marker for which putative pharmacoepigenetic relationships can be elucidated using available data.

The Genetics of DNA Methylation Consortium (GoDMC) is an international collaboration that aggregated data from >30,000 study participants to provide associations between genetic variants and DNAm sites in the general population known (Min et al., 2021). Epigenome-wide association studies (EWAS) characterize the association between DNAm and phenotypic outcomes such as aging and smoking, but also pharmacogenomic (PGx) outcomes. Over 2,500 such studies are hosted on the EWAS catalog (Battram et al., 2022), and the EWAS Atlas (Xiong et al., 2022), and the GTEx project (The GTEX Consortium, 2020). To further elucidate the pathway from DNAm to phenotype, association studies between methylation and gene expression can provide insightful information (Ruiz-Arenas et al., 2022), as do omics quantitative trait loci (QTL) data (The GTEX Consortium, 2020). In Table 1, we present publicly available resources that provide summary statistics of these molecular associations that could help identify causal pathways in the context of pharmacoepigenetics as subsequently discussed for the anti-psychotic compound clozapine (section "Synthesizing available pharmacoepigenetic data to investigate pharmacoepigenetic interactions").

In section "Synthesizing available pharmacoepigenetic data to investigate pharmacoepigenetic interactions," we illustrate the use of these resources to examine the epigenetics of clozapine. There are several relevant resources. dMEM, or the database of Epigenetic Modifiers, maintains the genomic information of about 167 epigenetic cancer target modifiers and proteins including DNAm and histone modification and microRNAs (Singh Nanda et al., 2016). Consortia such as NIH Roadmap Epigenomics and the

**Table 1.** Public resources with quantitative molecular interaction information that directly or indirectly involve DNAm

| Interaction | Interaction entities | Alternative interaction Name | Resources | Resources – References | Clozapine example | Clozapine example – References |
|---|---|---|---|---|---|---|
| 1 | Genetic variant – PGx outcome | PGx GWAS | GWAS Catalog PGRN-RIKEN | Buniello et al., 2019 | rs2472297 C > T clozapine plasma concentration ($P = 4.35 \times 10^{-10}$) | Pardiñas et al., 2019 |
| 2 | Genetic variant – methylation status | mQTL | GoDMC | Min et al., 2021 | rs2472297 C > T cg13570656 ($P = 4.73 \times 10^{-32}$); cg17852385 ($P = 7.79 \times 10^{-32}$); cg01359532 ($P = 5.03 \times 10^{-75}$) | Min et al., 2021 |
| 3 | Methylation status – PGx outcome | EWAS | EWAS Catalog EWAS Atlas EWASdb | Battram et al., 2022; Xiong et al., 2022; Liu et al., 2018 | NA | NA |
| 4 | Methylation status – gene expression | eQTM | Helix Project Human Kidney meQTL Atlas | Ruiz-Arenas et al., 2022; Liu et al., 2022 | Liver: cg14503537-CYP1A2 ($P = 1.2 \times 10^{-3}$); Blood/Liver cg13570656 – CYP1A1 (n.s.); cg13570656 – CYP1A2 (n.s.); cg17852385 – CYP1A1 (n.s.); cg17852385 – CYP1A2 (n.s.); cg01359532 – CYP1A1 (n.s.); cg01359532 – CYP1A2 (n.s.) | Bonder et al., 2014; Ruiz-Arenas et al., 2022 |
| 5 | Genetic variant – omics levels | Expression QTL Protein QTL Metabolite QTL | GTEx Portal OMICSCIENCE | The GTEx Consortium, 2020; Lotta et al., 2021 | rs2472297 C > T – CYP1A1 (n.s.) rs2472297 C > T – CYP1A2 (n.s) | The GTEx Consortium, 2020 |

*Note:* Resources to query interaction information (1–5) as displayed in Figure 2 are outlined for genetic variants, PGx outcome (e.g., differential drug response, ADR), DNAm at CpG sites, gene expression, and omics levels. Each interaction involves two of these entities and to facilitate navigation, we explicitly name them (i.e., genetic variant – DNAm level) while also mentioning the common name of the intended analysis (i.e., mQTL). Information about the rs2472297-*CYP1A1/CYP1A2* – clozapine example is provided when available. This list of resources is not exhaustive as we focused on large databases with user-friendly interfaces. As such, references supporting the clozapine examples also stem from other resources.

Encyclopedia of DNA Elements (ENCODE) provide data portals containing thousands of sequencing-based genome-wide epigenetic datasets (Fingerman et al., 2011; Luo et al., 2020). Another epigenetic resource, Epimap, combines 10,000 epigenomic maps across 800 samples, which annotate chromatin states, high-resolution enhancers, enhancer modules, upstream regulators, and downstream target genes (Boix et al., 2021).

Genome browsers such as the Washington University Epigenome Browser, offer a visualization platform that integrates consortia data for browsing and downloading (Zhou et al., 2015). The UCSC genome browser offers a GTEx track, which allows users to examine epigenetic variation and gene expression in a variety of human tissues (Kent et al., 2002; Navarro Gonzalez et al., 2021).

Aside from databases examining epigenetic markers, there are databases that provide drug-response information. The PharmGKB provides curated drug label annotations, clinical guideline annotations, FDA annotations, genetic variant annotations, curated pathways, and annotations for hundreds of drugs (Whirl-Carrillo et al., 2021). DGIdb (Drug–Gene Interaction database) provides information on drug–gene interactions and druggable genes from publications, databases, and other web-based sources (Freshour et al., 2021). DrugBank offers chemical, pharmacological, and pharmaceutical data with comprehensive drug target information (Wishart et al., 2018). Additionally, the human enhancer drug database (HEDD) integrates epigenetic drug datasets obtained from laboratory experiments and manually curated information. HEDD incorporates five kinds of datasets: (1) drug, (2) target, (3) disease, (4) high-throughput, and (5) complex datasets (Qi et al., 2016). Pharmacogenomic variation nomenclature is standardized within the PharmVar consortium as a centralized pharmacogene variation data and pharmacogenomic nomenclature repository (Gaedigk et al., 2021).

### Synthesizing available pharmacoepigenetic data to investigate pharmacoepigenetic interactions

Clozapine is commonly prescribed for schizophrenia. There is a wealth of information, which can be assembled about it from existing epigenetic and PGx resources. We illustrate how pharmacoepigenetic mechanisms can be hypothesized by integrating different datasets (Figure 2 and Table 1).

Clozapine has high interindividual differences in plasma clozapine concentration at a given dose and the risk of serious adverse drug reactions (ADR) at high concentrations can make its use challenging (Molden, 2021). A recent PGx genome-wide association study (GWAS) ($N = 2,989$) identified a single genetic variant (rs2472297) associated with plasma clozapine concentration located between *CYP1A1* and *CYP1A2* (interaction 1, Figure 2).

Consulting the GoDMC methylation QTL (mQTL) resource, we associated this SNP to three CpG sites, two (cg13570656, cg17852385) located in the proximity of the transcription start site of *CYP1A1* and one in the *CYP1A2* intron 5–6 (cg01359532; interaction 2). To the best of our knowledge, no epigenome-wide association study (EWAS) on clozapine plasma concentration has been reported, which could relate methylation status to observed interindividual concentrations (interaction 3). Previous studies provided evidence for inverse correlations between DNAm and *CYP1A2* mRNA levels in liver (Ghotbi et al., 2009;

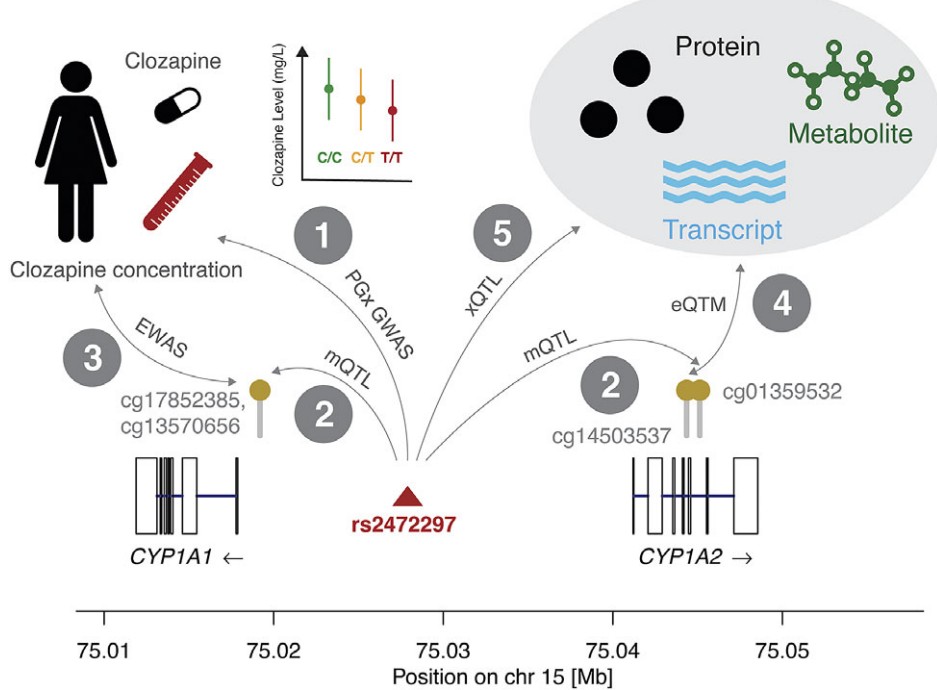

**Figure 2.** Combining molecular interaction resources can detect putative causal mechanisms that determine differential drug responses because of DNAm. We show interactions between genetic variants (SNPs), methylation status at CpG sites, drug response, expression levels, and other omics measures. In interaction 1, we report the association between the reduction in clozapine concentration and the minor allele of rs2472297 (Pardiñas et al., 2019). In interaction 2, we present the CpG sites in vicinity of the *CYP1A1* and *CYP1A2* genes whose methylation levels are under the genetic influence of rs2472297 (GoDMC mQTL study). This suggests that epigenetic mechanisms may affect clozapine concentration. Interaction 3 indicates the association results that could be expected from an EWAS on clozapine concentration, however, such data is currently not available for this compound. Interaction 4 represents the link between methylation and expression levels (eQTM) that could support the role of *CYP1A1* or *CYP1A2* as mediators in this hypothetical epigenetic mechanism. Interaction 5 represents genetic associations to omics data such as mRNA expression, protein levels, and metabolite levels, which could further provide mechanistic insights and elucidate downstream effects of methylation on clozapine concentration through other omics layers. This figure is accompanied by Table 1 with public resources to query quantitative information corresponding to these interactions. The molecular mechanism depicted here is based on the significant effect of rs2472297 on clozapine metabolite plasma concentration (Pardiñas et al., 2019). As outlined above, support for other interactions is often missing and we detail the degree of evidence in Table 1.

Bonder et al., 2014), among which is an inverse correlation between the intronic cg14503537 and *CYP1A2* (interaction 4) (Bonder et al., 2014). However, no significant correlation between any of the previously listed mQTL CpG sites and *CYP1A1/CYP1A2* was found in either liver or blood (Bonder et al., 2014; Ruiz-Arenas et al., 2022). Likewise, no expression QTL (eQTL) data involving these two genes was reported for rs2472297 in the GTEx project (interaction 5) (The GTEX Consortium, 2020).

While the data may suggest a relationship between rs2472297 and clozapine concentration through DNAm, the information is too sparse to exclude horizontal pleiotropy (i.e., the SNP affecting DNAm and clozapine concentration independently). However, DNAm may be a consequence of the lower gene expression due to the proximal SNP. Given the absence of a genetic association with expression levels, further studies would be necessary to corroborate this hypothetical mechanism. Even this simple example with clozapine demonstrates the complex space of possible direct and indirect pathways, including forward and reverse directionalities, and warrants caution when integrating and interpreting molecular associations.

### Synthesizing available pharmacoepigenetic data to investigate the DNAm landscape of CYP genes

*CYPs* play a major role in the metabolism of a large fraction of drugs and have notable variance in their expression, explained by both genetic and nongenetic factors. Studies previously discussed

(section "Epigenetic variation impacts nonepigenetic drug response") identify methylation near *CYP* promoter regions as a possible cause for differential drug response. We searched the GoDMC database to assess the DNAm landscape in 10 major *CYP* genes involved in drug metabolism (Min et al., 2021).

GoDMC provides *cis* and *trans*-mQTL information based on the analysis of 420,509 DNAm sites. We restricted our search to DNAm sites located 50kB up- and downstream of the *CYP* gene of interest and downloaded available data including the average DNAm level of these sites in the general population, mQTLs in *cis* influencing their DNAm levels, the minor allele frequency (MAF) of these mQTLs, and the estimated heritability of DNAm levels based on *cis*-mQTLs. In Table 2, we summarized this information. The full query results can be found in Supplementary Table S1.

DNAm levels in CYP genes range from 1.8% up to 97.9%. As a comparison, genome-wide DNAm levels are reported on average 52% across the ~420,000 tested CpG sites. However, only 21% of the CpG sites are in transcription factor-binding regions (Min et al., 2021). Across the CYP genes, the frequency and location of DNAm sites with respect to the transcript exons are gene-specific (Figure 3). Seven out of the 10 assessed genes (*CYP1A1, CYP2B6, CYP2C19, CYP2D6, CYP3A4, CYP3A5, and CYP4F2*) have at least one DNAm site in the promoter region whereas only intronic and/or exonic DNAm sites were observed for *CYP1A2* and *CYP2C9*.

We focused on *CYP*-DNAm sites that are under genetic control with heritability ranging from <0.1% up to 97%. Genetic variants

**Table 2.** Summary of DNAm variation for 10 *CYP* genes

| CYP gene | Chromosome | Number of DNAm sites | Mean DNAm level (min–max) | mQTL MAF (min–max) | *cis*-heritability (min–max) |
|---|---|---|---|---|---|
| CYP1A1 | 15 | 14 | 0.036–0.816 | 0.020–0.438 | 0.001–0.106 |
| CYP1A2 | 15 | 23 | 0.036–0.979 | 0.012–0.438 | 0.001–0.595 |
| CYP2B6 | 19 | 3 | 0.380–0.842 | 0.016–0.427 | 0.006–0.319 |
| CYP2C19 | 10 | 3 | 0.686–0.832 | 0.010–0.422 | 0.018–0.102 |
| CYP2C8 | 10 | 1 | 0.756 | 0.130 | 0.010 |
| CYP2C9 | 10 | 1 | 0.756 | 0.130 | 0.010 |
| CYP2D6 | 22 | 20 | 0.018–0.902 | 0.015–0.484 | 0.002–0.337 |
| CYP3A4 | 7 | 2 | 0.707–0.752 | 0.024–0.035 | 0.010–0.062 |
| CYP3A5 | 7 | 12 | 0.022–0.903 | 0.010–0.316 | 0.002–0.324 |
| CYP4F2 | 19 | 13 | 0.106–0.874 | 0.009–0.492 | 0.003–0.965 |

*Note:* For each gene, we include the chromosome, number of DNAm sites found 50 kB up- and downstream the gene boundaries, minimum and maximum mean DNAm levels, maximum mQTL minor allele frequency (MAF), and the minimum and maximum DNAm *cis*-heritability values. The mean DNAm level indicates the average methylation status of a given DNAm site in the studied population. The MAF corresponds to the MAF of the top mQTL (mQTL most significantly associated with that DNAm site) and the DNAm *cis*-heritability is the extent of DNAm variation explained by genetic variation in proximity of the gene which we derived by summing up the explained variance of independent *cis*-mQTLs. Only DNAm sites available in the GoDMC resource (whole blood) are reported with the study population being of European ancestry. Full query results with detailed mQTL association information can be found in Supplementary Table S1.

influencing these DNAm levels were found to be both rare (MAF < 5%) and very common (MAF > 40%). GoDMC shows that genetic variants influence 45% of the assessed DNAm sites. This suggests that *CYP* genes could harbor additional DNAm sites likely to be under environmental control. The DNAm data we are presenting here was derived from whole blood in European participants and differences in the liver, the most relevant tissue for CYP enzymes, as well as population-specific differences, are expected. While further research is needed to attribute DNAm sites to PGx effects and determine whether epigenetic marks are functional intermediates or consequences of differential gene activity, this qualitative assessment may help prioritizing candidate genes to conduct further research. For example, *CYP2D6* metabolizes several common drugs and is polymorphic in epigenetic profiles in some populations (Qi et al., 2020). Understanding the epigenetic landscape of the gene, including which genes are under genetic control may allow scientists to better prioritize which of these sites to pursue with biological experiments.

## Challenges in pharmacoepigenetics

Our current understanding of pharmacoepigenetics paired with available resources can generate hypotheses about interactions between epigenetics and classical nonepigenetic drug response. However, some of the information necessary to reconstruct or predict pharmacoepigenetic relationships is not yet available, as seen in the clozapine example.

Challenges in the field go beyond unavailable data. Epigenetic mechanisms are important for cancer progression, yet, studying pharmacoepigenetics in the context of cancer is challenging. In cancer cells, it is difficult to identify population-level genetic or epigenetic differences because the cancer epigenome is different from the host's somatic epigenome and cancer cells are rapidly evolving. Furthermore, in the setting of active treatment, it is challenging to track which markers are basal to the patient tissue, are cancer-specific, or are a result of a pharmacological action. The tissue-specificity of epigenetic signals also means that obtaining disease-relevant samples from patients via blood draw is often not

possible, limiting the current clinical applications of epigenetic profiles. Nevertheless, epigenetics marks affect cancer drug response and prognosis. Therefore, understanding these relationships remains an important area of research that is difficult to examine at the population and epigenome level of analysis.

Establishing causal relationships between changes in epigenetic state and a given phenotype is not only challenging in cancer. Environmental factors, disease states, and drugs can affect epigenetic markers, and likewise, marker levels may alter drug response and disease prognosis. Additionally, epigenetic states are dynamic and change throughout an individual's lifetime. As such, determining the cause and consequence of observed associations between epigenetic markers, medical conditions, and drug responses is not always straight-forward. In some cases, twin studies are used to determine causality, but much remains to be uncovered regarding the directionality of the relationship between epigenetics and drug response (Bell and Spector, 2012).

To navigate these complexities and disentangle observed associations, causal inference methods such as Mendelian randomization (MR) techniques can consolidate forward or reverse causality (Porcu et al., 2021a). MR makes use of genetic instrumental variables and has been successful in identifying putative causal relationships between gene expression and complex traits using eQTL and GWAS. Analogous application to mQTL data can reveal DNAm-to-trait and even DNAm-to-gene expression-to-trait pathways (Sadler et al., 2022; Figure 1). Equivalently, MR can be used to test whether altered methylation levels are responsible for observed PGx effects by leveraging PGx GWAS (Auwerx et al., 2022). While EWASs of the PGx phenotype of interest can provide evidence of putative mechanisms, these results should be interpreted with caution considering the challenges associated with determining causality. Indeed, it was found that observed DNAm-to-trait and gene expression-to-trait correlations were more likely to arise due to reverse causality (i.e., trait-induced) (Min et al., 2021; Porcu et al., 2021a).

As more population-level epigenetic data emerges, there is an opportunity to elucidate potential pharmacoepigenetic interactions from publicly available data by (1) identifying pharmacogenes with unexplained variability, (2) locating key epigenetic biomarkers near

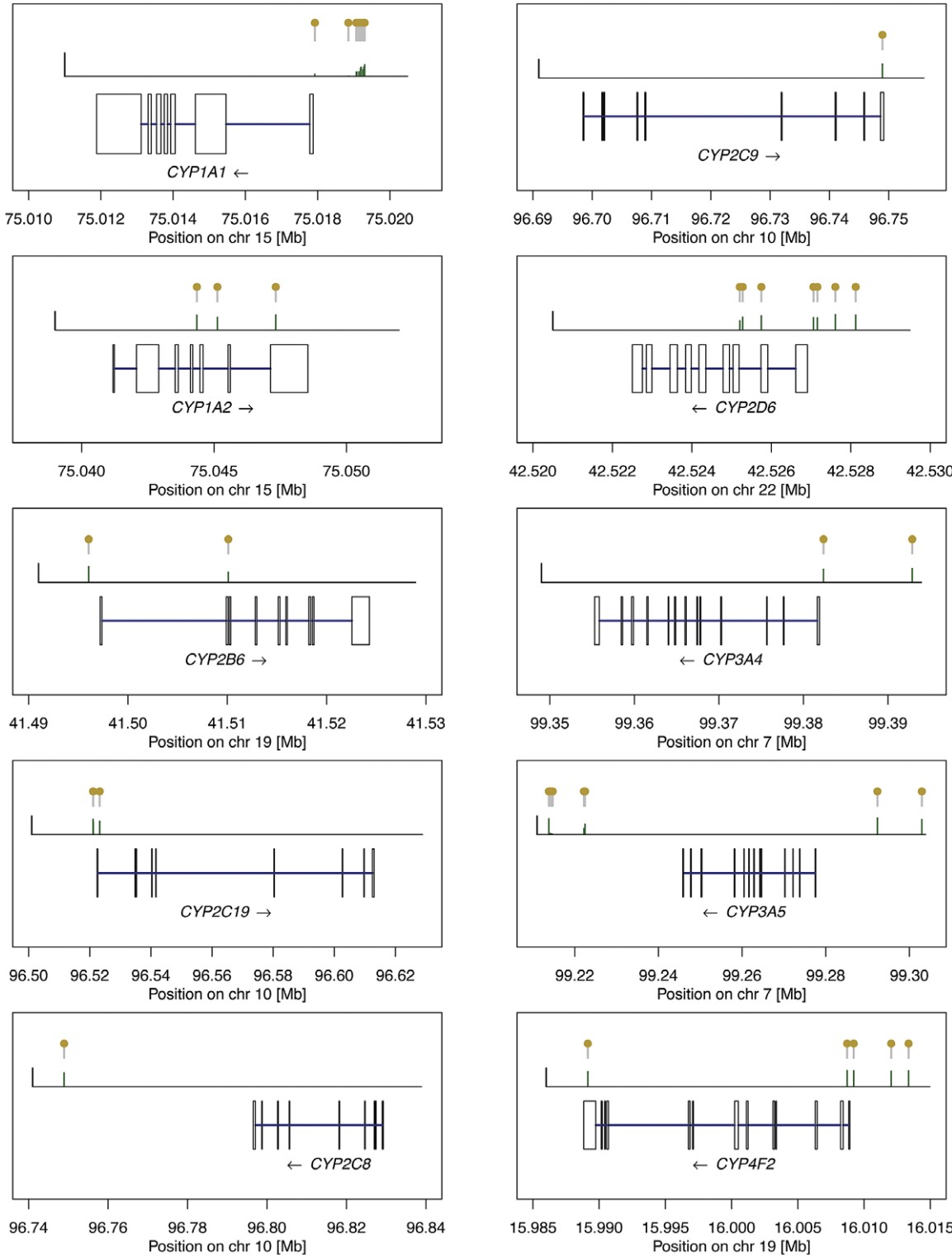

**Figure 3.** Visualization of DNAm profiles for 10 CYP genes. This data was generated from public resources to demonstrate the wealth of epigenetic information available about important drug metabolism genes. Each box represents one CYP gene with the name and strand orientation. The exon/intron architecture is outlined and aligned to the position on their respective chromosomes. DNAm sites are drawn above (gray bars with yellow dots) relative to their location on the gene together with their average DNAm level (green bars). The height of the left black bar indicates a DNAm level of 1 (i.e., 100% methylated). DNAm sites very close to each other may appear as a single bar and for visualization purposes, DNAm sites distant to the gene body were omitted. DNAm site positions and DNAm level information are from the GoDMC resource (whole blood).

the gene or regulatory region, and (3) referencing epigenetic databases to hypothesize relationships (such as those outlined in Figure 1) between biomarkers like DNAm and phenotypes such as drug response. It will be critical to validate such predicted interactions via biological and clinical experiments to generate a deeper understanding of epigenetic effects and drug response. While there is a great deal of information already available, more is required to fully examine pharmacoepigenetic influences on drug

response. Additional pharmacoepigenetic studies on classical nonepigenetic drugs both in vitro and with patient populations would help address these gaps. Employing novel computational approaches to identify or predict pharmacoepigenetic relationships paired with biological validation allows us to fully materialize the promise of pharmacoepigenetics as a powerful tool for understanding biological mechanisms and developing effective interventions.

**Open peer review.** To view the open peer review materials for this article, please visit http://doi.org/10.1017/pcm.2023.6.

**Supplementary material.** The supplementary material for this article can be found at https://doi.org/10.1017/pcm.2023.6.

**Financial support.** D.A.S. was supported by the Stanford Biochemistry Department, the NSF Graduate Research Fellowship Program ID: 2019286895. M.C.S. was supported by the Swiss National Science Foundation (310030-189147) and would like to thank the Fulbright Program for funding her research stay at Stanford University. R.B.A. was supported by NIH GM102365 HG010615 and the Chan-Zuckerberg Biohub.

**Competing interest.** The authors have no conflict of interest to disclose.

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
