## [Reviewer Report]

*Comments to Author*: In the manuscript by Smith and colleagues, the authors discuss how epigenetic markers i) can be indicative of disease, ii) can improve patient diagnosis and prognosis and iii) impact drug response. Unfortunately, the work remains overall superficial. Epigenetic mechanisms and principles are not discussed. Effects and important distinctions between histone and DNA modifications are not even mentioned and the work focusses exclusively on DNA methylation. Similarly, other epigenetic DNA marks, such as hydroxymethylation that could be of relevance, particularly in the liver, are not mentioned.

Other specific comments:

1) Since the focus of the work should be on pharmacoepigenetics, the first two aspects should be either shortened or the connection to pharmacokinetics, drug response or toxicity (i.e. not only to disease) should be made more clear.

2) The manuscript is more of a commentary or opinion piece rather than a classical review of pharmacoepigenetics. I would suggest to change the manuscript type accordingly and to fully commit to one or the other format. I.e. reduce speculative elements and original analyses if the work is supposed to be a review. Also, the structure with a separate Discussion section is rather unusual for a review article.

3) Abstract: “Pharmacogenetics, the study of how interindividual genetic differences affect drug response does not explain all observed variance in drug response.” – In fact, pharmacogenetics only explains a relatively small fraction of the interindividual variability. The authors are recommended to either narrow the scope to “observed heritable variance in drug response” or to rephrase the sentence.

4) miRNAs should not be included in such a review. I am aware that miRNAs have been included in similar works before, but this should not justify its inclusion, as the definition of epigenetics becomes much too broad. See PMID 29339796.

5) “Genetic variation accounts up to 95% of all drug response variance, although often the percentage is below 50%” – This seems like a gross overestimation. There are individual probe substrates for which the heritable, which is not the same as genetic, variation has been shown to approximate 95% (see PMID 30684656), but most common estimates suggest that “only” around 10-30% of variability is explained by genetic factors.

6) The authors should more critically discuss limitations of the clinical application of epigenetic biomarkers, such as the tissue-specificity of epigenetic signals which entails that epi signatures other than those in the peripheral blood remain largely inaccessible.

7) Figure 2 seems to be identical with the graphical abstract.

8) “While the data may suggest a relationship between rs2472297 and clozapine concentration through DNAm, the information is too sparse to exclude horizontal pleiotropy (i.e., the SNP affecting DNAm and clozapine concentration independently).” – A different explanation could be that DNAm is a consequence of the lower 1A2 expression due to the proximal SNP, i.e. that DNAm does not have a functional role in this case and is merely a bystander.

9) In the context of the comment before, the possibility that most epigenetic marks are not functional intermediates, but rather only the consequence, i.e. markers, of gene activity modulation should be transparently discussed.

10) Section 2.6: “While further research is needed to attribute DNAm sites to PGx effects, this qualitative assessment may help prioritizing candidate genes to conduct further research.” – Could the authors please explain how the presented data helps in the prioritization of sites for further study? Maybe examples could be provided of sites that, in the opinion of the authors, might be more likely to have functional impacts on CYP expression.

---

## [Reviewer Report]

*Comments to Author*: In this paper the authors present some of the features of the current state of knowledge of pharmacoepigenetics and some of the associated promises and challenges.The manuscript is good, but I have the following comments and queries:

1. Abstract, line 27: I think "account for the unexplained variance" should be changed to : "account for some of the unexplained variance"

2. Abstract,line 28: "powerful" should be replaced by"suitable" or "good"

3. Abstract, lines 30 and 33: I have not come across the terms "reverse pharmacoepigenetics" and "forward pharmacoepigenetics" before. Can the authors provide a reference for these terms (the reference need not be inserted in the abstract). If the authors cannot find a reference for the terms, may be the terms can be deleted from the manuscript.

4. Introduction, line 50: "they" should be replaced by "he or she responds"

5. Introduction, line 67: "consist" should be replaced by "include"

6.Section 2.1, lines 123-126: the data on mental illness has not been confirmed. Can you make the sentence a little more tentative?

7.Section 3.1, line 351: "can be" needs to be changed to "are"

8. Section 3.1, line 361: DNAm-to-to-trait" is wrong. Please amend.

9. Section 3.1,line 366: "care" should read as "caution"

10.Section 2.3, line 325:Please insert "the" before liver.

11. Discussion section, first paragraph, lines 336-339: The use of "However" twice does not sound good. Please change.

12. Reference section: The way of referencing by the authors is not uniform. Please make it uniform.

---

## [Editor Report]

*Comments to Author*: Thank you for this comprehensive review. Please consider carefully the comments of both reviewers, in particular reviewer 1. However, miRNA may be still included in the manuscript, but it should be mentioned that by definition, miRNAs are not always considered as an epigenetic factor.

---

## [Reviewer Report]

*Comments to Author*: The manuscript has improved after revision. However I do find two papers cited in the text in the reference list: 1. Page 4, line 88 - Furtado et al, 2019

2. Page 4, line 89 - Licht, 2021

Please make changes in the reference list.

---

## [Editor Report]

*Comments to Author*: 

Please add the full reference of Furtado et al, 2019 to the list of references,